# High-Content RNAi Phenotypic Screening Unveils the Involvement of Human Ubiquitin-Related Enzymes in Late Cytokinesis

**DOI:** 10.3390/cells11233862

**Published:** 2022-11-30

**Authors:** Mikaël Boullé, Laurianne Davignon, Keïs Nabhane Saïd Halidi, Salomé Guez, Emilie Giraud, Marcel Hollenstein, Fabrice Agou

**Affiliations:** 1Center for Technological Resources and Research (C2RT), Department of Structural Biology and Chemistry, Chemogenomic and Biological Screening Core Facility, Institut Pasteur, Université Paris Cité, CNRS UMR 3523, F-75015 Paris, France; 2Laboratory for Bioorganic Chemistry of Nucleic Acids, Department of Structural Biology and Chemistry, Institut Pasteur, Université Paris Cité, CNRS UMR 3523, F-75015 Paris, France; 3Collège Doctoral, Sorbonne Université, F-75005 Paris, France

**Keywords:** cytokinesis, CEP55, ubiquitin signaling, RNAi, image analysis tool, high-content screening

## Abstract

CEP55 is a central regulator of late cytokinesis and is overexpressed in numerous cancers. Its post-translationally controlled recruitment to the midbody is crucial to the structural coordination of the abscission sequence. Our recent evidence that CEP55 contains two ubiquitin-binding domains was the first structural and functional link between ubiquitin signaling and ESCRT-mediated severing of the intercellular bridge. So far, high-content screens focusing on cytokinesis have used multinucleation as the endpoint readout. Here, we report an automated image-based detection method of intercellular bridges, which we applied to further our understanding of late cytokinetic signaling by performing an RNAi screen of ubiquitin ligases and deubiquitinases. A secondary validation confirmed four candidate genes, i.e., *LNX2*, *NEURL*, *UCHL1* and *RNF157*, whose downregulation variably affects interconnected phenotypes related to CEP55 and its UBDs, as follows: decreased recruitment of CEP55 to the midbody, increased number of midbody remnants per cell, and increased frequency of intercellular bridges or multinucleation events. This brings into question the Notch-dependent or independent contributions of LNX2 and NEURL proteins to late cytokinesis. Similarly, the role of UCHL1 in autophagy could link its function with the fate of midbody remnants. Beyond the biological interest, this high-content screening approach could also be used to isolate anticancer drugs that act by impairing cytokinesis and CEP55 functions.

## 1. Introduction

The achievement of complete cell division requires the separation of the nuclear and the cytoplasmic content [1]. However, the understanding of mitosis is far more advanced than the knowledge about cytokinesis [2,3]. For a long time, it has been technically easier to study the separation of chromosomes than to follow the different steps of cytokinesis. Besides, cancer investigations have mainly focused on the consequences of pathological modifications of any form in nucleic acid sequences, rather than on changes in functional–structural or cellular phenotypes [4]. The relative dependence between mitosis and cytokinesis varies between cell types, but, canonically, cytokinesis begins with the spindle assembly checkpoint, at the end of which the anaphase-promoting complex/cyclosome complex (APC/C) degrades cyclin B, thereby inactivating the cyclin-dependent kinase 1 (CDK1) [5,6]. The kinases aurora B and polo-like kinase 1 (PLK1) also contribute to the initiation of the central spindle assembly [5]. Early cytokinesis corresponds to the contraction of the actomyosin ring perpendicularly to the central spindle microtubules. Then, the stabilisation of the nascent intercellular bridge and the maturation of its central midbody part during late cytokinesis lead to the cytokinetic abscission [6].

Centrosomal protein 55 (CEP55) is a central regulator during this late stage in human cells, where it participates in midbody structuration and ESCRT machinery recruitment, hence contributing to both the abscission checkpoint and the final disruption of the intercellular bridge [7,8]. Interestingly, CEP55 persists after abscission in midbody remnants, which are degraded either by autophagy or after engulfment reminiscent of phagocytosis, and possibly mediate differential cell fate by asymmetric inheritance [9,10,11,12]. CEP55 is stabilised during mitosis by a series of post-translational modifications. It has been shown that CDK1 and extracellular signal-regulated kinase 2 (ERK2) phosphorylate the S425 and S428 serines, which favours the cis-trans isomerisation of the neighbouring prolines by peptidyl–prolyl cis-trans isomerase NIMA-interacting 1 (PIN1) [13,14]. These modifications are taking place in a proline-rich linker between two CEP55 ubiquitin-binding domains (UBDs) [15]. We have recently reported that the C-terminal ubiquitin-binding zinc finger (UBZ) is necessary and sufficient to recruit CEP55 to the midbody, whilst the NEMO–optineurin–ABIN domain (NOA) contributes to post-recruitment abscission [15]. The phosphorylation by PLK1 of serine 436 delays the recruitment of CEP55 until the inhibitor of apoptosis stimulating protein of p53/protein phosphatase 1 complex (iASPP-PP1) dephosphorylates the same serine [16,17]. 

The role of several phosphorylations is well characterised in cytokinesis, whereas the contribution of the ubiquitin signaling is not well understood [5,18,19,20,21]. Ubiquitin is grafted as a monomer or a chain on substrate proteins to modify their cellular compartment and their assembly or, conversely, to promote their proteasomal degradation [22]. Even if other ubiquitin linkages exist, degradative K48 can be roughly distinguished from non-degradative polyubiquitin M1 and K63 chains [23]. We showed recently that the latter are specifically recognised by CEP55′s UBDs, thereby connecting the structural and functional importance of these domains with the cell biology of cytokinesis [15]. Nevertheless, we do not know the ubiquitinated substrate(s) recognised by CEP55, although it is generally accepted that its recruitment depends on mitotic kinesin-like protein 1 (MKLP1) [13]. Ubiquitination is performed by the consecutive action of E1 ubiquitin-activating enzymes, E2 ubiquitin-conjugating enzymes and E3 ubiquitin ligases, respectively catalysing the activation, conjugation and the ligation of a ubiquitin monomer, usually on a lysin or the initial methionine of a substrate [24,25]. The reverse reaction is catalysed by deubiquitinases (DUB) [26]. Few ubiquitin ligases and DUBs have been observed at the intercellular bridge or at the midbody, without much, if any, functional description of their structural roles [20,21,27,28,29,30,31,32,33]. However, a clear description of the ubiquitin signaling during late cytokinesis has not emerged yet. The quantification of the phenotypes induced by the depletion of CEP55 or the invalidation of its UBDs has given us the opportunity to understand more about this signaling. These interconnected phenotypes include an increased frequency of intercellular bridges and of multinucleation events and a decreased recruitment of CEP55 to the midbody, as well as an increased number of midbody remnants per cell. In this report, we screened for ubiquitin-related enzymes involved in late cytokinesis by RNA interference.

Over time, forward and reverse genetics have given insightful information about cytokinesis in different organisms and cell systems [34,35,36,37]. Besides, the emergence of RNA interference has brought a more detailed understanding of the different essential genes involved in cytokinesis [38,39,40,41,42,43]. This may be more challenging with CRISPR-Cas9-mediated knockout methods where complete disruption of essential genes can affect cell viability [44]. Nevertheless, most, if not all, the screens focusing on cytokinesis have used multinucleation as the endpoint readout, thereby only detecting the genes whose downregulation results in the absence of bridge formation or in the collapse of it [38,39,40,41,42,43]. Indeed, live cell imaging has confirmed that the knockdown of some involved genes does not always lead to multinucleation, but to an increased duration or length of the intercellular bridge [38,45,46,47,48]. Since no method to detect cytokinetic bridges in high-content screening and fixed conditions was available, we decided to code such an image-based tool. After validation, we used it to screen for ubiquitin-related enzymes by automatically measuring the frequency of cytokinetic bridges.

## 2. Materials and Methods

### 2.1. Cell Lines and Cell Culture

Human HeLa cells (American Type Culture Collection, Manassas, VA, USA, CCL2) were cultivated in DMEM supplemented with 10% heat inactivated FBS and penicillin/streptomycin, 100 units/mL and 100 mg/mL, respectively.

### 2.2. siRNAs and Screening Bank

To optimise and test the automated detection of intercellular bridges, we downregulated *CEP55* by separately using several different siRNAs: siGENOME siRNAs D-006893-05-0005 (GGAGAAGAAUGCUUAUCAA), D-006893-06-0005 (UAACACAGUUGGAAUCCUU), D-006893-07-0005 (GAAGAGAAUGA UAUUGCUA), D-006893-08-0005 (GCGAUCUGCUUGUCCAUGU) from Dharmacon (Lafayette, CO, USA) and the SI02653021 (CAGGUUAUUGCUAAUGGGUUA) from Qiagen (Venlo, The Netherlands). During this development, scrambled control siRNAs from Dharmacon were employed: siGENOME non-targeting siRNAs D-001210-02-05 (UAAGGCUAUGAAGAGAUAC), D-001210-03-05 (AUGUAUUGGCCUGUAUUAG), D-001210-04-05 (AUGAACGUGAAUUGCUCAA), D-001210-05-05 (UGGUUUACAUGUCGACUAA); siGENOME non-targeting siRNA pool D-001206-13-20 (UAGCGACUAAACACAUCAA, UAAGGCUAUGAAGAGAUAC, AUGUAUUGGCCUGUAUUAG, AUGAACGUGAAUUGCUCAA); ON-TARGETplus non-targeting siRNA pool D-001810-10-20 (UGGUUUACAUGUCGACUAA, UGGUUUACAUGUUGUGUGA, UGGUUUACAUGUUUUCUGA, UGGUUUACAUG UUUUCCUA). Some experimental control conditions were mock-transfected. The human siRNA siGENOME library targeting genes coding for ubiquitin ligases and deubiquitinases was obtained from Dharmacon (GU-006205-E2-01). The bank was resuspended in siRNA buffer (B-002000-UB-100, Dharmacon) according to instructions given by the manufacturer. During the screening, the negative scrambled control siRNA pool (D-001206-13-20) and the positive controls against *CEP55* (D-006893-05-0005, D-006893-06-0005, D-006893-07-0005, D-006893-08-0005) were randomised on each plate. Validation of the candidate genes was performed with the same individual siGENOME siRNAs targeting respective screening hits: *LNX2* (D-007164-01-0005, CCAAGUGGCUCUUCAUAAA), *UCHL1* (D-004309-01-0005, UAGAUGACAAGGUGA AUUU), *MYSM1* (D-005905-01-0005, GAAGAGAACUGUACAAAGG), *HECW2* (D-007192-01-0005, GCAGAGAUCUAACUCCAUA), *RNF7* (D-006907-02-0005, UCUUAGA UGUCAAGCUGAA), *RNF157* (D-022965-01-0005, UGAGAAGCCUGGUCAAUAU), *WDR59* (D-022683-01-0005, GAGCUGAAGUGUUGAAGUU), *TRIM25* (D-006585-01-0005, GACCGCAGCUGCACAAGAA), *RNF13* (D-006944-01-0005, GAAACUUCCUGUA CAUAAA), *VPS8* (D-023668-01-0005, GCAAUAAGCUCCUUGUAUA), *NEURL* (D-016715-01-0005, CCACAAGGCUGUCAAGAGG), all ordered from Dharmacon. The controls during validation of the candidates were the negative scrambled siRNA pool (D-001206-13-20) and a siRNA against *CEP55* (D-006893-05-0005).

### 2.3. Screening and Transfection

The bank of siRNAs targeting the genes encoding ubiquitin ligases and deubiquitinases was randomly distributed in 96-well plates, as well as the negative and positive controls per plate. The plate borders were not used to avoid edge effects. Three thousand HeLa cells in 50 µL medium were reverse transfected with 25 nM final siRNA concentration and 0.1 µL DharmaFECT 1 transfection reagent (T-2001-02, Dharmacon). Forty-eight hours post-transfection the cells were fixed with 4% PFA. During optimisation of the automated detection and during validation of the candidates, the reverse transfection was performed in 24-wp on glass coverslips and the volumes adapted proportionally to reach the same concentrations. To validate the siRNAs and verify the downregulation of candidate genes, the reverse transfection was performed in 6-wp and the volumes adapted proportionally to reach the same concentrations.

### 2.4. Immunofluorescence and Imaging

The transfected and fixed cells were treated with 50 mM NH_4_Cl to reduce autofluorescence before permeabilisation with 0.2% Triton X-100. Once blocked with PBS 1× – 1% BSA, primary staining was performed for 1 h at room temperature with the following antibodies: 1:2000 mouse monoclonal anti-β-tubulin (clone TUB 2.1), Sigma-Aldrich (T4026), St. Louis, MO, USA; 1:300 mouse monoclonal anti-CEP55 (clone B8), Santa Cruz Biotechnology (sc-374051), Dallas, TX, USA; 1:1000 rabbit polyclonal anti-MKLP1 (N-19), Santa Cruz Biotechnology (sc-867), USA). Following washes in PBS 1x, incubation was performed under the same conditions with the following secondary antibodies: 1:1000 goat polyclonal anti-mouse IgG1 coupled with AF488 (ThermoFisher Scientific A-21121, Waltham, MA, USA); 1:1000 goat polyclonal anti-mouse IgG2a coupled with AF546 (ThermoFisher Scientific A-21133, USA); 1:1000 goat polyclonal anti-rabbit coupled with AF647 (ThermoFisher Scientific A-21245, USA). After counterstaining with DAPI, the plates were imaged on an Opera Phenix High-Content Screening system (PerkinElmer, Waltham, MA, USA), 12 images per well at 20× *g* magnification. Alternatively, coverslips were mounted on slides using Mowiol and images acquired with an Axio Imager Z1 (Carl Zeiss, Oberkochen, Germany) at the same magnification.

### 2.5. RT-qPCR

Total RNA from transfected HeLa cells was extracted using a Nucleospin RNA kit (Macherey Nagel 740955.50, Duren, Germany) according to the manufacturer’s protocol. A SYBR Green-based real-time PCR assay (Luna Universal One-Step RT-qPCR Kit, New England Biolabs E3005S, USA) was performed in a final volume of 5 μL per reaction in white ultraAmp 384-well PCR plates using a QuantStudio 6 Flex real-time PC system (Applied Biosystems 4485689, Waltham, MA, USA). Briefly, 2 μL of RNA was added to 2.5 μL of 2X Luna Universal One-Step Reaction Mix and primers (Eurofins Genomics, Nantes, France) at a final concentration of 0.4 μM (sequences available on request). The PCR program included a reverse transcription step at 55 °C for 10 min, followed by 40 cycles of denaturation at 95 °C for 10 s and extension at 60 °C for 1 min. SYBR Green fluorescent emission was measured at the end of the elongation step. Subsequently, a melting curve program was applied with a continuous fluorescent measurement starting at 60 °C and ending at 95 °C (ramping rate of 0.05 °C.s^−1^). Crossing point values (Cp) were determined by the second derivative maximum method in QuantStudio software. For result normalisation, the following control genes were tested with the Normfinder program [49]: *Actin B* (beta actin), *B2M* (beta-2 microglobulin), *GAPDH* (glyceraldehyde-3-phosphate dehydrogenase), *PPIA* (peptidylprolyl isomerase A), *HPRT* (hypoxanthine–guanine phosphoribosyltransferase). *PPIA* was selected as the most stable reference gene for the Hela cells. The relative RNA quantities were calculated as follows: (E_target_)^ΔCp(target)^/(E*_ppia_*)^ΔCp(*PPIA*)^, with E the PCR efficiency of each primer pair and ΔCp the Cp difference between control and sample for each gene [50].

### 2.6. Software Development

Our tool to automatically detect intercellular bridges was written in Java as a plugin within the Icy image analysis software [51]. The source code is available in GitLab and Zenodo. The plugin can be downloaded from the Icy website, as described in the data availability section. Identification of the midbody, the nucleus and the cytoplasm by detection of the respective signal is required to classify a midbody as part of an intercellular bridge or as remnant. The logic behind this plugin is explained in the results and the discussion.

### 2.7. Image Analysis and Quantification

All the analyses were executed with the Icy image analysis software, with which DAPI-stained nuclei and MKLP1 spots can be automatically detected by the HK-means method. During the development of the image analysis plugin, intercellular bridges were first detected by eye, then the results were compared to those obtained with different versions of the program. An average of one thousand cells was analysed per condition during this initial step. The optimal plugin, whose code is available, was applied for screening, where an average of two hundred and fifty cells were analysed per condition. During the screen, the frequency F of intercellular bridges was estimated by F = I/(N − I) × 100, where the number of cells is calculated by subtracting the number of cytokinetic bridges I to the number of nuclei N in the condition. The validation of the candidates was performed by manually and blindly counting the frequency of cytokinetic bridges in an average of two hundred and fifty cells per replicate. The number of multinucleated cells counted manually was also subtracted from the number of nuclei during the validation of the candidates, since the multinucleation phenotype was measured concomitantly. The frequency F′ of intercellular bridges then became F′ = I/(N − I − M) × 100, where M is the number of multinucleated cells. Similarly, the frequency F″ of multinucleated cells was estimated by F″ = M/(N − I − M) × 100. The recruitment of CEP55 to the midbody was evaluated by measuring, within each MKLP1 spot, the average intensity ratio of the CEP55 to the MKLP1 signal. The number of midbody remnants was estimated by subtracting the number of intercellular bridges from the number of MKLP1 spots. Then, the number of midbody remnants per cell was calculated and expressed as a percentage.

### 2.8. Representation and Statistical Analysis

Data were represented and statistics were calculated with R. The Spearman’s rank correlation coefficient was estimated to evaluate the performance of the automated detection of intercellular bridges. To estimate the statistical strength of an effect during the screen, the z-score was calculated for each tested siRNA by z = (x − µ)/σ, where µ and σ are, respectively, the average frequency and the standard deviation of intercellular bridges in all the tested wells from the plate where this siRNA was transfected, and where x is the frequency of cytokinetic bridges in this specific well. Hits were selected based on the z-score and the frequency of intercellular bridges. The validation of candidate genes with different CEP55-related phenotypes was verified by adjusted Wilcoxon–Mann–Whitney tests, comparing the distribution of the replicates for each hit with those obtained with the non-targeting scrambled siRNA pool. Images and graphical representations were assembled with Inkscape software.

## 3. Results

### 3.1. Development of Automated Detection of Cytokinetic Bridges

Taking benefits from the environment of the Icy image analysis software, we developed a plugin written in Java to automatically detect intercellular bridges. To this aim, we first had to detect the different types of biological object before interrogating their relationships. The preliminary step was to determine optimal and robust immunofluorescence staining conditions for each relevant type of object. DAPI and β-tubulin are frequently used to stain nuclei and cytoplasms, respectively. As part of the Centralspindlin complex, MKLP1 is one of the earliest elements of the midbody and mediates the recruitment of CEP55, which highlights its relevance as a marker for the midbody. Staining and imaging conditions were easily calibrated and are summarised in the methods (Figure 1).

To detect nuclei and midbodies, we applied HK-Means, combining hierarchical clustering with the K-Means methods [52]. This segmentation method executes an N-class thresholding based on a K-Means classification of pixels from the histogram of their intensities in the image, before extracting regions of interest (ROI) in a bottom-up manner following parameters defined by the user, i.e., minimum and maximum size of the clustered object as well as its minimum intensity. This allowed us to easily count the number of nuclei and the number of midbodies per field of view (FOV) and per condition. The boundaries of the detected nuclei were refined by progressively fitting the contour of each nuclear ROI to the nuclear shape, following an active contouring method based on intensity gradient and edge detection [53]. Each midbody spot was expanded by a constant user-defined number of pixels in order to draw a larger spot, from which the initial one was subtracted, hence generating a midbody ring. Given that the cytoplasms of future daughter cells remain connected until abscission, we decided to avoid their segmentation. We simply generated a single region of interest per image encompassing all detected cytoplasms. This cytoplasmic ROI was drawn by automated thresholding using K-Means, where the number of intensity classes to determine the minimum positivity threshold is user-defined (Figure 1a,b). 

Once delineated, we interrogated the relationships between nuclear, midbody and cytoplasmic ROIs to estimate the number of cytokinetic bridges. Midbodies were either classified as remnants or as intercellular cytokinetic bridges. The ring surrounding each midbody spot was examined to establish how many times it crossed the β-tubulin signal and if it was intersecting with any DAPI-positive ROI. Midbody rings with a single intersection or without any interaction with the cytoplasmic ROI were classified as midbody remnants. When at least two intersections were counted, the midbody was considered as part of a cytokinetic bridge. If the ring was intersecting with or inscribed in a nucleus, the midbody was classified as remnant (Figure 1a). In terms of output, the plugin overlays all the detected ROIs on each respective image of a sequence (Figure 1b). Per field of view and per condition, the number of nuclei, the number of midbodies and the number of cytokinetic bridges, as well as the criteria for the classification of each midbody, are exported. The plugin and the code are accessible, as described in the data availability section.

To evaluate the accuracy of the automated detection, we compared its results with those of a preliminary blind manual detection in a set of 68 conditions from four different experiments, with an average of 1000 analysed cells for each condition (Figure 1c). In each one, HeLa cells were mock-treated or transfected with one of several anti-*CEP55* or scrambled non-targeting siRNAs at 25 nM. The number of detected cytokinetic bridges was subtracted from the number of nuclei to estimate the number of cells. The frequency of cytokinetic bridges was then calculated per condition and expressed as a percentage. Manually, the frequency of cytokinetic bridges in mock-transfected conditions and in those treated with scrambled siRNA measured below 10%. Upon downregulation of *CEP55* with an siRNA, the cytokinesis frequency consistently measured above 10%. This is the reason why we chose 10% as the threshold to evaluate the accuracy of the classification by the automated image analysis tool. Each software version was similarly tested until we were satisfied with the accuracy of the detection. The current and herein described version of the software showed the best performance, with a Spearman correlation coefficient of 0.87. The linear correlation was confirmed by a visually rather linear distribution of the data and a significant result of the correlation test, with *p* < 10^−15^. With R^2^ = 0.75, three-quarters of the observed variation can be explained by the correlation. The parameters of the affine function summarising the correlation curve were the slope a = 0.88 and the translation b = 0.21. By applying the threshold of 10% to the results of the automated detection, we were able to evaluate the performance of the classification. In brief, the accuracy of the software was equal to 0.93. The precision and recall values were 1 and 0.78 respectively. Taken together, the accuracy of the automated detection of intercellular cytokinetic bridges allowed us to screen for ubiquitin-related enzymes in late cytokinesis.

In Appendix A, we show the performance of the detection of cytokinetic bridges by the software in WT and *CEP55* KO U2OS cell lines. We have already published data on these cell lines. The software could easily classify the cell type based on the frequency of cytokinetic bridges. However, the detection was not optimal, as the slope of the regression was close to 0.5. Firstly, this was due to the fact that the cytokinetic bridges in these cells are not as long as in HeLa cells. Secondly, the environment of each individual cell is rather crowded, since U2OS cells grow in small colonies. Thus, we decided to work with HeLa cells, in which the effect of gene downregulation was more efficiently detected. Nevertheless, these results have increased our confidence in the robustness of our tool.

### 3.2. Screening of Ubiquitin Ligases and Deubiquitinases Involved in Late Cytokinesis

To address a relevant biological question with our image analysis tool, we screened a library of siRNAs targeting ubiquitin ligases or DUBs to detect their involvement in late cytokinetic steps (Figure 2). We applied the same experimental conditions as during optimisation of the automated detection, i.e., a 48 h-long incubation after transfection of 25 nM siRNA. The position of the tested siRNAs was randomised on 96-well plates, each of them including controls. Positive controls were treated with four different siRNAs against *CEP55*. Negative controls were either mock-treated or transfected with scrambled non-targeting siRNAs (Figure 3a). Since we did not consistently reach a positive z’-factor between positive and negative controls during the screening setup, we only used these conditions as experimental controls for each plate and to evaluate the consistency between plates. As planned, wells at the edge of each plate were avoided to prevent border effects. The staining and analysis conditions were with the same as those used during optimisation.

Six hundred and seventy-one genes encoding either a ubiquitin ligase or a deubiquitinase were targeted across all thirteen plates, with at least six positive or negative controls per plate. An average of 250 cells were analysed per condition. The frequency of intercellular cytokinetic bridges was the endpoint readout to estimate the occurrence of late cytokinetic events. Across plates, the downregulation of *CEP55* by the different positive control siRNAs led to an increase in the frequency of intercellular bridges to 17.48 +/−4.01% (mean +/− SD). In contrast, the average frequency of cytokinetic bridges was 6.04 +/− 1.90% in the wells treated with scrambled non-targeting siRNAs and 6.29 +/− 1.73% in the mock-treated conditions. We did not observe any plate effect. An 11% positivity threshold of cytokinesis frequency was set, above which hits were selected (Figure 3b). We calculated a z-score per tested siRNA based on the mean and the standard deviation of all the tested siRNAs on the same plate (Appendix A). This score and the available literature have guided us to narrow the selection down to eleven hits, whose downregulation increased the frequency of cytokinetic bridges. These hits are listed by the name of the targeted gene, associated with the respective observed frequency of intercellular bridges, as follows: *UCHL1*, 16.51%; *NEURL*, 16.29%; *TRIM25*, 15.60%; *WDR59*, 13.85%; *RNF7*, 13.81; *HECW2*, 12.64%; *LNX2*, 12.20%; *MYSM1*, 11.45%, *RNF157*, 11.40%; *RNF13*, 11.31%; *VPS8*, 11.11%. In total, 1.64% of the targeted genes were considered positive after this initial screen. The results obtained with all the tested siRNAs from the library are available in Appendix A.

### 3.3. Validation of the Screening Based on CEP55-Related Phenotypes

As described in our recent publication, the downregulation of *CEP55* leads to four different phenotypes (Figure 4). The increase in the frequency of intercellular bridges and in multinucleation results from the inhibition of late cytokinesis, due to the lack of CEP55 at the midbody [7,13]. This absence also results in the increase in the number of midbody remnants per cell, possibly due to the lack of midbody recycling, either through autophagy or lysosomal degradation after engulfment [9,10,11]. These interconnected phenotypes appear when CEP55′s ubiquitin-binding domains are debilitated. Therefore, we expected phenocopies when downregulating some ubiquitin-related enzymes. As part of the validation of the primary screen, we blindly evaluated the presence of these same phenotypes under downregulation of the different hit genes with respective siRNAs (Figure 5). For each hit, six biological replicates were performed in HeLa cells under the same conditions of downregulation as during the screen, i.e., 48 h post-transfection with 25 nM siRNA. An average of 250 cells were analysed per replicate. Downregulation of *CEP55* was established as the positive control, while the negative scrambled control was the reference for the statistical tests.

Targeting the ligand of Numb-protein X 2 (*LNX2*) or the neuralised E3 ubiquitin protein ligase 1 (*NEURL*) resulted in a reproducible significant increase in the frequency of cytokinetic bridges (Figure 5a). More multinucleated cells were observed when knocking down the ring finger protein 157 (*RNF157*) or *NEURL* (Figure 5b). The downregulation of the ring finger protein 13 (*RNF13*), *RNF157*, *LNX2* or the ubiquitin C-terminal Hydrolase 1 (*UCHL1*) variably reduced the intensity of the CEP55 signal at the midbody (Figure 5c). When *RNF157* or LNX2 was knocked down, the number of midbody remnants decreased relative to the number of cells, whereas the number of remnants per cell increased after downregulation of *UCHL1* (Figure 5d).

While extending the frame of the primary screen through the consideration of new readouts, we decided to preserve reasonable stringency and to select only candidates whose downregulation alters at least two of the CEP55-related phenotypes. Thus, these data allowed the validation of *LNX2*, *NEURL*, *UCHL1* and *RNF157*. Complementary results shown in the Appendix A validate the efficient and specific downregulation of these candidates as well as *CEP55* by siRNA. Despite our careful demonstration, caution should be exercised during the exploration of each remaining candidate gene until cross-validation has been performed using an alternative method to RNA interference, as we used only one siRNA per gene. To illustrate the direct effect of their downregulation on the recruitment of CEP55 to the midbody, representative images are shown in Figure 6. Altogether, our results help the detection of late cytokinetic events and contribute to the understanding of the ubiquitin signaling linked to CEP55 and cytokinetic abscission.

## 4. Discussion

Since previous cytokinetic screens relied on the detection of multinucleated cells, our first technical aim was to enable the endpoint detection of intercellular cytokinetic bridges. The constant improvement of mathematical functions to detect standard biological objects in microscopic images has allowed us to write a plugin within the environment of the Icy image analysis software. This plugin first recognises the stained nuclei, cytoplasms and midbodies, before classifying the latter as remnants or as part of cytokinetic bridges. The performance of the automated detection in HeLa and U2OS cells gives us confidence in our ability to detect and classify the same structures in other fixed cell lines or even in primary cells. This tool will become very useful to identify active compounds or any biochemical modifier that blocks the late cytokinetic events leading to abscission. However, this method may underestimate the impact of some perturbations in cytokinesis. Indeed, inhibiting cytokinesis frequently leads to the absence of a cytokinetic bridge or its collapse, both leading to multinucleation and probably to cell death through incompletely characterised molecular mechanisms [45,54,55]. Nevertheless, the reliable detection of intercellular bridges might be useful to perform high-content screening of late cytokinetic events. This would balance the low throughput of live imaging, which is usually employed to study the duration of cytokinesis, especially the last steps leading to abscission [38,48]. Here, we used this approach to screen for ubiquitin ligases and DUBs involved at this stage of the cell cycle.

The screen was carried out despite the statistically limited difference between positive and negative controls, but the setup, execution and analysis were eased by our technical development. The threshold of positivity for hits as well as the z-scores gave us reasonable confidence in our screen. Based on the knowledge accumulated in our previous publication about CEP55′s UBDs, we decided to validate the hits whose downregulation alters at least two CEP55-related phenotypes [15]. While the downregulation of *CEP55* reached nearly 100%, the downregulation of the gene candidates was slightly less efficient, with a minimum of 80% (Appendix A).

However, *NEURL* and *LNX2* increased the frequency of intercellular bridges upon knockdown. In *D. melanogaster*, NEUR ubiquitinates the Delta and Jagged ligands to promote its endocytosis, hence inducing the Notch signal in *trans* [56,57]. In mammalians, the contribution of its homolog NEURL to this function in the Notch pathway appears to be complemented by MIB1, whose downregulation did not alter the frequency of cytokinetic bridges in our screen [56,57]. It should be noted that the lack of phenotypic effect observed with MIB1 could be due to the presence of several functional and redundant genes in mammalian cells. LNX2 is also known to target Numb for proteasomal degradation, hence influencing the Notch signaling in *cis* [58,59]. However, the role of Notch in cytokinesis has never been highlighted, despite clearly modulating the fate of daughter cells during asymmetric division [60]. While a putative link could be established between late cytokinesis and Notch signaling, other hypotheses need to be formulated, mainly because targeting MIB1 did not show any effect in our screen [61]. NEURL may have other functions, for which the ligands are unknown [62]. The downregulation of *NEURL* increased both the frequency of cytokinetic bridges and multinucleated cells, which is clearly in favour of it having a role in cytokinesis. In the case of LNX2, the number of specific protein substrates for ubiquitination is steadily growing. Both LNX2 and its paralog LNX1 trigger degradative ubiquitin signal, which does not explain how the depletion of LNX2 could directly decrease the recruitment of CEP55 to the midbody [63,64]. It is possible that CEP55 is not recruited to the midbody because of its disorganisation, probably concomitant with the degradation of its protein components. LNX1 and LNX2 are structurally similar and display common protein substrates for ubiquitination [59,63,65]. Recently, proteomic studies have evidenced their potential substrates such as MID1 and MID2, as well as KIF14 [66]. The first two are involved in early cytokinesis, whereas the latter participates in the stabilisation of the central spindle, hence in the structuration of the intercellular cytokinetic bridge [31,67]. 

The results about UCHL1 are noteworthy, though not directly linked to cytokinetic bridges. Indeed, CEP55 could be associated with the ubiquitinated target leading to the recycling of the midbody. The downregulation of *UCHL1* led to a reduced intensity of CEP55 on midbodies and to an increased number of midbody remnants per cell. Both effects may suggest a lack of ubiquitin chain edited by UCHL1 on a substrate of the midbody. The recycling of midbodies may also be altered by directly disturbing autophagy [68]. Indeed, it has been reported that CEP55 interferes with the formation of LC3-positive autophagosomes [69]. As for RNF157, the two observed effects, i.e., the reduced intensity of CEP55 at the midbody and the increased frequency of multinucleated cells, may be more difficult to explain. It is known that the downregulation of *RNF157* induces apoptosis in neuronal cells through a mechanism involving Fe65, and that it participates in the editing of dendrites [70]. RNF157 has also been described as a downstream effector of the PI3K and MAPK pathways [71]. In our study, the total number of cells was significantly decreased during downregulation of *RNF157*, possibly by apoptosis. Interestingly, the downregulation of *CEP55* in zebrafish leads to massive apoptosis of neural structures, concomitantly with a reduced AKT activation [72]. In addition, CEP55 is phosphorylated by the ERK2 kinase [13]. Nevertheless, these theoretical analogies linking CEP55 with RNF157 require further experimentation, especially since the hydranencephaly of the MARCH syndrome is associated with a defect in the recruitment of CEP55 to the midbody [73].

To conclude, we repeat that we should use our tool for screening. Nowadays, cytokinesis remains less understood than mitosis, although cytoplasmic division contributes equally to genome integrity. Specifically, the regulated rupture of the intercellular bridge needs to be understood so that techniques can be devised to block it in various types of cancer. As reported in the discussion section of our previous article, we are still looking for elements that condensate the midbody to a state of liquid–liquid phase transition, and we have previously observed ubiquitin at the midbody. Therefore, we have used our bioimage analysis tool to look for elements of the ubiquitin signaling system whose downregulation affects cytokinesis. Each candidate gene should be further investigated using an alternative method to siRNA knockdown, such as CRISPR interference or a knockout method. Even though we did not deeply characterise the functional role of each candidate gene that results from our siRNA screening, we strongly believe that the replication of cytokinetic defects and the co-occurrence of other CEP55-related phenotypes upon the downregulation of candidate genes in multiple validation experiments strengthen our approach. Indeed, our discovery of functional ubiquitin-binding domains in CEP55 was the main reason why we decided to perform a late cytokinesis screen to search for ubiquitin-related enzymes that could be associated with the function of CEP55 during and after cytokinesis. In the future, it may also be important to investigate the role of these cytokinetic elements in healthy primary cells.

## Figures and Tables

**Figure 1 cells-11-03862-f001:**
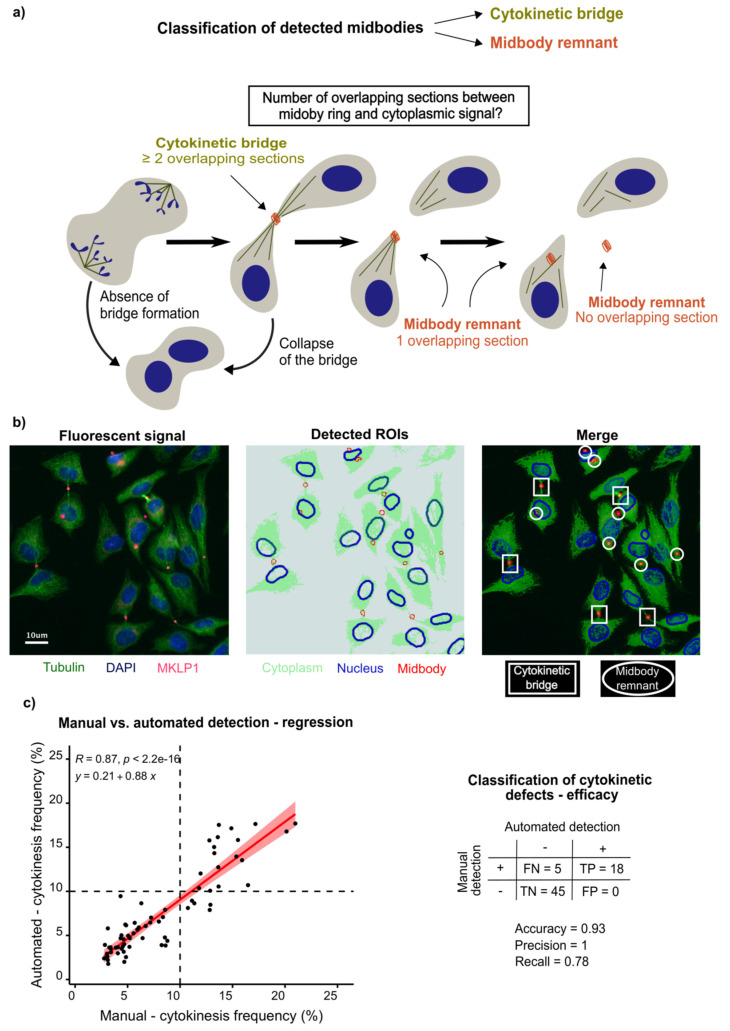
Development of automated detection of intercellular bridges. (**a**) Unlike the observation of multinucleation, the detection of intercellular bridges is a more appropriate phenotype to study late cytokinesis. A midbody present at the intercellular bridge is connected to each future daughter cell, whereas a midbody remnant is observed within one cytoplasm or outside cells. To classify each midbody, we detected the number of overlapping sections between each drawn surrounding midbody ring and the cytoplasmic ROI. (**b**) All three necessary elements to classify midbodies were delineated by automated detection of respective signals in HeLa cells: DAPI for nuclei (blue), β-tubulin for cytoplasms (green) and MKLP1 for midbodies (red). The image on the right is the overlay of the merged signals and the detected regions of interest, i.e., left and middle image respectively. The colour code is maintained between the detected signals and the represented ROIs. On the overlay, examples of detected midbodies classified as cytokinetic bridges (rectangles) or midbodies (ovals) are highlighted. (**c**) Performance of the bioimage analysis tool in HeLa cells. Left, the frequencies of cytokinetic bridges from a blind manual detection of intercellular bridges were plotted against those obtained automatically by the program. The dots represent 68 experimental conditions from four different experiments. In each condition, cells were transfected either with an siRNA targeting *CEP55*, a non-targeting scrambled siRNA or mock-transfected. The correlation between the results of the two methods was appreciated by a Spearman correlation test. The vertical dashed line corresponds to a threshold which separated the mock-transfected and the conditions transfected with a scrambled siRNA from the conditions transfected with an siRNA targeting *CEP55*: blindly and manually measured, their cytokinesis frequency was always found exclusively below or above 10% respectively. The same threshold was applied for the automated detection of cytokinetic bridges and represented by a horizontal dashed line. Right, the efficacy of the classification was estimated based on these thresholds, while considering manual counting as the gold standard. The number of true (TP) or false (FP) positive and of true (TN) or false (FN) negative results of the classification allowed the calculation of performance parameters such as accuracy, precision and recall.

**Figure 2 cells-11-03862-f002:**
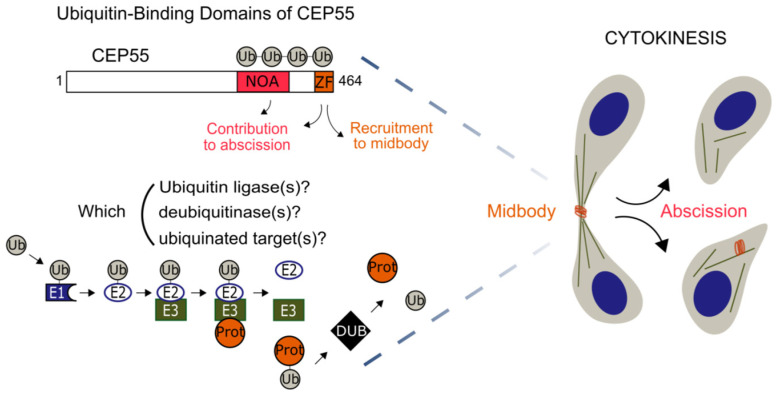
Descriptive diagram of the raised biological questions. CEP55 is a key actuator of late cytokinesis. The zinc finger (ZF) domain in CEP55 cargoes the protein to the midbody, whereas NOA contributes to the cytokinetic abscission. The presence of these two functional ubiquitin-binding domains in CEP55 structurally links the ubiquitin signaling with late cytokinesis. Ubiquitination requires three successive reactions named activation, conjugation and ligation, catalysed, respectively, by E1, E2 and E3 ubiquitin enzymes. Ubiquitin architectures are edited or withdrawn from substrates by deubiquitinases. Such molecular actors are not known regarding the recruitment of CEP55 to the midbody and its late cytokinetic function, i.e., abscission and recycling of the midbody remnants.

**Figure 3 cells-11-03862-f003:**
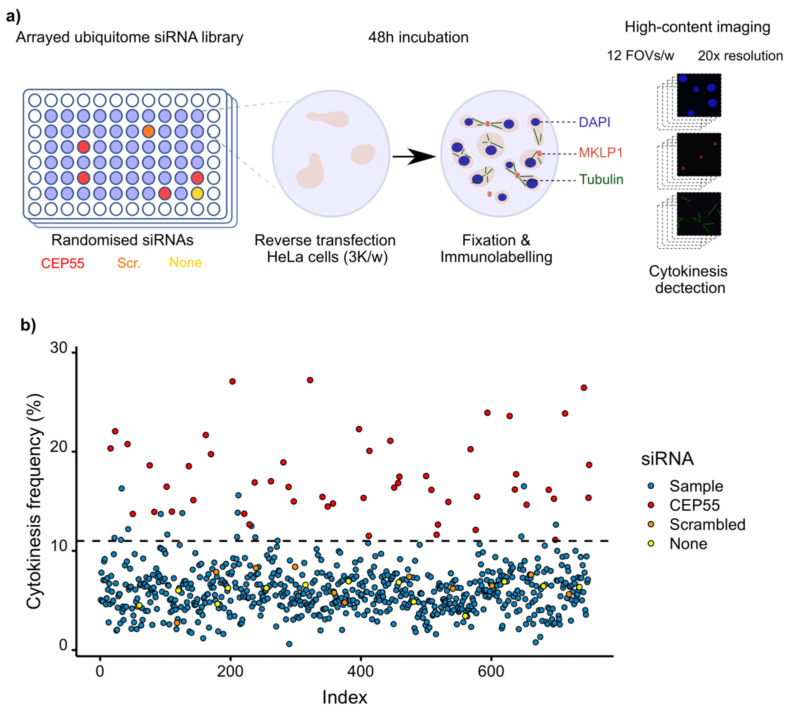
Overview of the screening execution. (**a**) Representation of the screening pipeline from the plate map until imaging. (**b**) The distribution of the screening results indexed according to their chronological order of acquisition does not show any plate effect. The dashed line represents the positivity threshold above which samples were considered as hits.

**Figure 4 cells-11-03862-f004:**
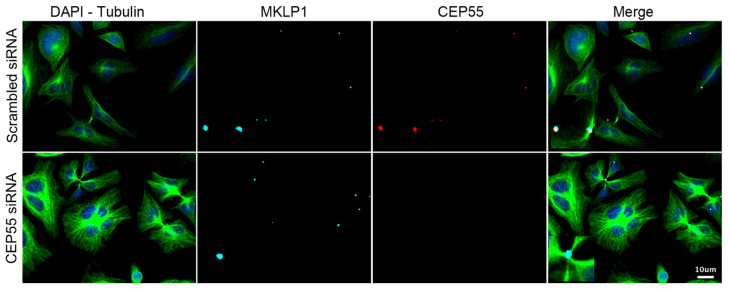
Phenotypes induced by the downregulation of *CEP55*. Immunofluorescence images of HeLa cells 48 h post-transfection either with 25 nM non-targeting scrambled or anti-*CEP55* siRNA. The identification of nuclei, cytoplasms and midbodies was enabled by staining with DAPI, anti-β-tubulin and anti-MKLP1 antibodies. The staining of CEP55 allows the evaluation of its expression and its recruitment to the midbody. Insets correspond to a 4-fold magnified view of a region in the respective image.

**Figure 5 cells-11-03862-f005:**
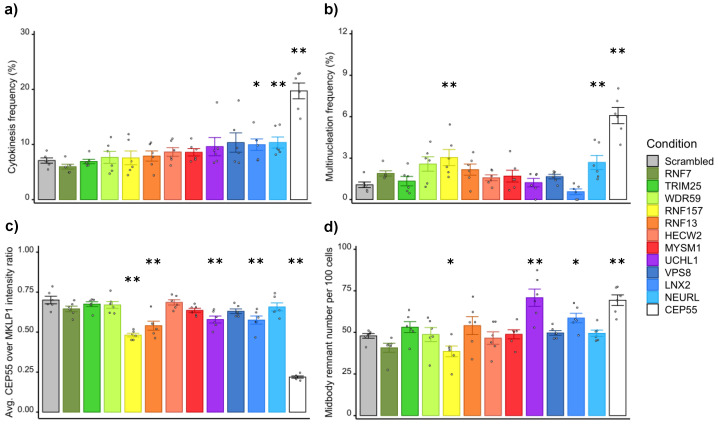
Validation of candidate genes by measurement of CEP55-related phenotypes. HeLa cells were transfected 48 h with 25 nM of siRNA against each selected and color-coded hit. (**a**) Frequency of cytokinetic bridges. (**b**) Frequency of multinucleated cells. (**c**) Average intensity ratio of CEP55 to MKLP1 signals per midbody. (**d**) Number of midbody remnants per 100 cells. Per condition, mean and standard error are represented as well as individual replicates. For each condition, six biological replicates were performed. The total number of cells and the total number of midbodies analysed in all replicates are respectively shown in (**a**,**c**). Wilcoxon–Mann–Whitney tests compare the distributions of the replicates, where the scrambled siRNA condition is the reference. * *p* < 0.05; ** *p* < 0.01.

**Figure 6 cells-11-03862-f006:**
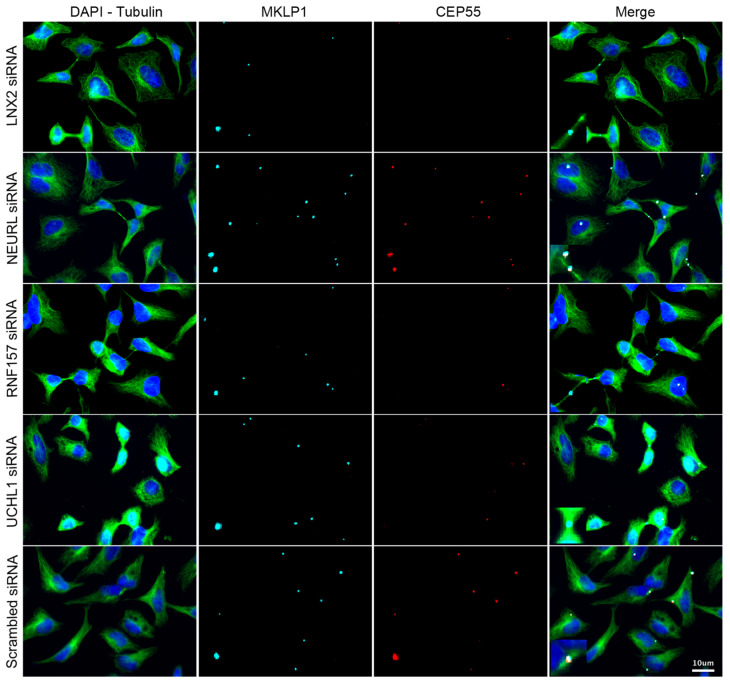
Representative phenotypes observed in validated candidates. Immunofluorescence images of HeLa cells 48 h post-transfection either with 25 nM non-targeting or anti-candidate siRNA. The identification of nuclei, cytoplasms and midbodies was enabled by staining with DAPI, anti-β-tubulin and anti-MKLP1 antibodies respectively. The staining of CEP55 allows the evaluation of its expression and its recruitment to the midbody. Insets correspond to a 4-fold magnified view of a region in the respective image.

## Data Availability

The source code is available in the GitLab (https://gitlab.pasteur.fr/pfccb/cytokinesis) and Zenodo (https://zenodo.org/record/5562194) repositories. and the plugin can be downloaded from the website of the open source Icy image analysis software (http://icy.bioimageanalysis.org/plugin/cytokinetic-bridge-detector).

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
