# Peer review of "High-Content RNAi Phenotypic Screening Unveils the Involvement of Human Ubiquitin-Related Enzymes in Late Cytokinesis"

_cells, 2022, doi:10.3390/cells11233862_

Round 1

Reviewer 1 Report (Previous Reviewer 2)

In this revised manuscript the authors have made several improvements.  First, they have utilized data from biological, rather than technical, replicates in the statistical analysis of the studies in Figure 5.  This greatly bolsters confidence in the image analysis pipeline to detect cytokinetic defects. In addition, they have included data showing that the tested oligos do effectively knock down the target transcripts. 

With regard to validation of the screen, the issue still remains that the same single siRNA oligo from the screen was used for validation of a hit (a single oligo is employed in the screen Table S1, and the same oligo is used in the validation screens as per the methods). While these oligos do knock down the target gene and behaved reproducibly in the secondary analyses, it can only be concluded that these oligos reproducibly produce an effect on cytokinesis that may be due to the gene of interest. As stated for the prior submission, it is standard practice for screens based on siRNA, shRNA, CRISRP to use multiple oligos/hairpins per gene to show that hits are caused by on-target effects and limit false positives. It is similarly standard practice to utilize multiple oligos for the target gene in biological analyses. At the least, a second oligo, independent of that used in the screen should be used for the validation studies to raise confidence that the genes identified are indeed playing roles in cytokinesis.  Ideally, proper validation would be performed to highlight these potentially interesting findings.  At the least, the authors need to clearly state this caveat to the identification of these genes and should soften the abstract accordingly.    

Author Response

We appreciate that the reviewer 1 found that we have improved our resubmitted manuscript. Nevertheless, the reviewer felt that we need to soften the biological conclusion of our work.

The aim of this study was to report a method which has brought some candidate genes potentially involved in late cytokinesis through the Ubiquitin signaling. We agree with the reviewer that each candidate will require further investigations and have accordingly moderated our manuscript regarding the validation of the candidate genes. Given the technical importance of this point, we realised that this part was poorly discussed in the manuscript, and we have now completed the method and discussion sections based on the reviewer’s suggestions.

All changes we have made in this new revised version are highlighted throughout the manuscript.

Reviewer 2 Report (Previous Reviewer 1)

The authors have sufficiently addressed the remaining points

Author Response

We thank the reviewer for having reviewed our article again.

This manuscript is a resubmission of an earlier submission. The following is a list of the peer review reports and author responses from that submission.

Round 1

Reviewer 1 Report

In this study, the authors report an automated image-based detection method for intercellular bridges as readout for cytokinesis and perform a RNAi screen of ubiquitin ligases and deubiquitinases to detect ubiquitin signalling-related regulators of cytokinesis. This is based on previous finding of the group that the cytokinesis factor CEP55 possess ubiquitin interaction motifs. A secondary validation of 11 primary hits confirmed the phenotype (increased cytokinesis frequency) for two candidates. For two other candidates the secondary validation showed other cytokinesis related phenotypes (multi-nucleation).

The screening method is valid and well explained (some minor points listed below). However, the manuscript stops after rescreening candidates and does not address their biological role in cytokinesis. Thus, the manuscript presents a screening approach, which is valid but conceptually and method-wise not overwhelming innovative but fails to answer the biological question, i.e. how does ubiquitin signalling (or at least the factors identified in the screen) contribute to cytokinesis.

Specific points:

1.) I recommend to present the results of the screening in plots of ranked z-score to illustrate tendencies and global deviations rather than in a single table. For the automated-manual comparison in addition the regression, I recommend to add the precision, recall and accuracy values.

2) The figure 6, and eventually the figure 4, as they are, do not add valuable information as it is difficult the see the MKLP1 and CEP55 spots. Magnified selections of the spots would clarify the message.

3) Screening and assays are all done in HeLa cells. It might be worth to include other cell line to show robustness of the screening method and probably as first step in generalizing the results for the hits.

Author Response

Dear reviewer,

Please find our answers attached.

Best regards

Reviewer 2 Report

In the present manuscript, the authors report the development of an imaging-based analysis platform with improved capabilities for detection of cytokinesis and alterations in this process that do not rely on the presence of multi-nucleation as a readout. They also report the use of this tool in an siRNA-based screen to identify components of the ubiquitin proteasome pathway in cytokinesis.  The development of the analysis platform is promising with several potential applications. However, the RNAi data is underdeveloped and unfortunately does not serve to demonstrate the utility and accuracy of the software.  In the absence of more extensive validation of the siRNA data, a demonstration of the software’s ability to identify known cytokinesis regulators in a screen setting would be more useful.  These concerns and opportunities to improve the manuscript for the reader are described below.   

Overall Concerns:

1.       The software appears to do a very good job of detecting cytokinesis events in comparison to manual classification.  However, this comparison could be better explained.  The meaning of “…cytokinesis detection (%)” on the axes of the graph could be better explained.  It seems this means the frequency of cytokinesis in the sample, but nature of the samples analyzed, and the 68 conditions used in the 4 experiments could be expanded upon to help the reader understand the power of the software. It would also be helpful to demonstrate that it can detect changes in cytokinesis caused by components of the cytokinesis machinery in addition to CEP55.

2.       There are several concerns with the siRNA data.  First, the screen is performed with single oligos to each gene. Generally, such a screen would be performed with multiple oligos per gene and hits would be identified when multiple oligos per gene scoring as significant in the assay.  Second, validation of hits from the siRNA screen is performed with a single siRNA rather than multiple and it is the same oligo used in the screen. While this validates the ability of this oligo to score in the assay, it does not validate that knocking down the putative target gene effects cytokinesis. Third, there is similarly no examination of the ability of these oligos to knockdown the gene of interest.  Fourth, with regard to impacts on CEP55 recruitment, there is no determination that CEP55 levels are not altered by the siRNAs.  The validation screen in Fig 5 is described as being performed in 3 biological replicates consisting of 3 technical replicates.  It appears that the data for each technical replicate is plotted and used for statistical analysis, which is not appropriate as it artificially increases the number of samples from 3 to 9, thus skewing the results of the analysis. The conclusion that ubiquitin pathway components involved in cytokinesis have been identified is not clearly supported by the data.    

3.       In general, the manuscript is written in a way as to suggest that the screen identified proteins involved in recruitment of CEP55 to the midbody and modulate phenotypes caused by altering CEP55.  But CEP55 seems to be more the rationale for looking at components of the UPS and the data would be more accurately described as affecting cytokinesis. CEP55 also regulates cytokinesis, but these phenotypes can be caused by perturbing events that CEP55 may not be involved in.

Suggested improvements for clarity:

1.       The description of Fig 1A in the text is easier to understand than the terminology used in the figure – for example discussing midbodies in terms of overlap rather than the number of tubulin branches. The relationship of the “measured cytokinesis-related phenotypes” panel at the right of Fig 1A to either the figure or the screening analysis method is not clear.

2.       Figure 1B would benefit from an indicating example of the different classifications of the midbody ROIs.  

3.       In Fig 1B (and beyond) it would facilitate understanding of the reader to indicate what the different colors represent in the image itself rather than only in the legend.

4.       Fig 2 and legend discusses ubiquitin-binding domains, but it is not clear to the reader where these reside in CEP55.  The terms ZFCEP55 and NOACEP55 should be more clearly defined. The figure might also benefit from depicting an interaction between CEP55 and ubiquitin at the midbody to better integrate the two aspects of the figure.

5.       A text box blocks part of 3a. 3b and its description in the legend are hard to interpret.  Why are hits shown by an index of their chronological acquisition?

6.       Fig 4 and 6 should include a label to the left indicating that these are siRNAs. This image would also benefit from having a merge of the MKLP1 and CEP55 staining alone.  The four phenotypes associated with CEP55 down-regulation should be clearly indicated.  

7.       Fig 5 parts of panels A and B are obscured by text boxes. The y-axis label in D as “ratio (%)” is somewhat confusing. Should it be a ratio as in panel C?

Author Response

(The authors gave the same response as above.)

Reviewer 3 Report

The manuscript entitled "High-content RNAi phenotypic screening unveils the involvement of human Ubiquitin-related enzymes in late cytokinesis." has been submitted as a original article by Bulle et al.

The authors describe an automated image-based detection method for cytokinesis mutants that focuses on intercellular bridges instead of using nucleation as endpoint. The authors identify and characterize the ubiquitin-linked enzymes LNX2, NEURL, UCHL1 and RNF157 as possible new candidates involved in CEP55-related functions.

In general, the experimental design of the study is fine, the experiments are sound and the conclusions are supported by the presented data.

Minor ponts:

1) The authors should include the information on how many independent experiments their statistical analysis is based in each case.

2) Figure 1, Figure 3 and Figure 5 have some formatting issues, because the panel numbering (a-d) is doubled.

3) The manuscript text needs moderate English language editing.

Author Response

(The authors gave the same response as above.)

Round 2

Reviewer 2 Report

In this revised manuscript the authors have improved readability of the manuscript.  However, scientific concerns remain. 

1.       The issues with the RNAi screen remain.  The use of multiple siRNAs to each gene is used in screening to limit the number of false-positives, as several oligos for a specific gene must score in the assay to identify that gene as a hit.  The validation experiments only confirm the impact of the oligo and not the protein of interest. Silencing efficiency needs to be shown under the conditions of the current experiment.  Technical replicates cannot be used for statistical analysis. These data only indicate how well a particular experiment performed and have no biological information.  These data artificially increase the number of samples from 3 to 9, thus skewing the results of the analysis.

2.       While the involvement of CEP55 in cytokinesis does provide rationale for looking at the ubiquitin machinery, it remains that there is nothing linking any of the proteins with identified as potential regulators of cytokinesis with CEP55.